# Transfer Learning Approach to Vascular Permeability Changes in Brain Metastasis Post-Whole-Brain Radiotherapy

**DOI:** 10.3390/cancers15102703

**Published:** 2023-05-10

**Authors:** Chad A. Arledge, William N. Crowe, Lulu Wang, John Daniel Bourland, Umit Topaloglu, Amyn A. Habib, Dawen Zhao

**Affiliations:** 1Department of Biomedical Engineering, Wake Forest School of Medicine, Winston-Salem, NC 27157, USA; carledge@wakehealth.edu (C.A.A.);; 2Department of Engineering, Wake Forest University, Winston-Salem, NC 27101, USA; 3Department of Radiation Oncology, Wake Forest School of Medicine, Winston-Salem, NC 27157, USA; 4Department of Cancer Biology, Wake Forest School of Medicine, Winston-Salem, NC 27157, USA; 5Clinical and Translation Research Informatics Branch, National Cancer Institute, Rockville, MD 20850, USA; 6Department of Neurology, University of Texas Southwestern Medical Center and VA North Texas Medical Center, Dallas, TX 75390, USA

**Keywords:** transfer learning, convolutional neural network, dynamic contrast-enhanced MRI, glioblastoma, brain metastasis, whole-brain radiotherapy

## Abstract

**Simple Summary:**

Dynamic contrast-enhanced (DCE) MRI has become a quantitative standard for assessing vascular permeability and perfusion. However, conventional pharmacokinetic (PK) modeling in DCE MRI is complex and time-consuming for dynamic MR scans with thousands of pixels per image. We have previously developed a deep learning approach using convolutional neural networks (CNN) as an efficient and accurate tool for the generation of PK parameter maps from DCE MRI of glioblastoma (GBM) mice. In the present study, the utility of this approach is further established through transfer learning between GBM-trained CNN and whole-brain radiotherapy (WBRT)-treated brain metastasis (BM) mice.

**Abstract:**

The purpose of this study is to further validate the utility of our previously developed CNN in an alternative small animal model of BM through transfer learning. Unlike the glioma model, the BM mouse model develops multifocal intracranial metastases, including both contrast enhancing and non-enhancing lesions on DCE MRI, thus serving as an excellent brain tumor model to study tumor vascular permeability. Here, we conducted transfer learning by transferring the previously trained GBM CNN to DCE MRI datasets of BM mice. The CNN was re-trained to learn about the relationship between BM DCE images and target permeability maps extracted from the Extended Tofts Model (ETM). The transferred network was found to accurately predict BM permeability and presented with excellent spatial correlation with the target ETM PK maps. The CNN model was further tested in another cohort of BM mice treated with WBRT to assess vascular permeability changes induced via radiotherapy. The CNN detected significantly increased permeability parameter Ktrans in WBRT-treated tumors (*p* < 0.01), which was in good agreement with the target ETM PK maps. In conclusion, the proposed CNN can serve as an efficient and accurate tool for characterizing vascular permeability and treatment responses in small animal brain tumor models.

## 1. Introduction

Brain metastasis (BM) is the most common intracranial tumor among adults, encompassing nearly 80% of all brain tumors [1,2,3]. Even with aggressive standard-of-care treatment, including surgical resection, chemotherapy, stereotactic radiosurgery, and/or whole-brain radiotherapy (WBRT), the prognosis is extremely poor, with a median survival of only 4–6 months [4,5,6]. In particular, conventional chemotherapeutic agents often fail when they are used to treat brain tumors due to many BM lesions retaining a partially intact blood–tumor barrier (BTB) [2,5,7]. As such, novel strategies aimed at enhancing the delivery of these chemotherapeutic agents are of paramount interest. It has been documented that WBRT may be able to disrupt the BTB to enhance the accessibility of chemotherapeutics to brain tumor parenchyma by increasing intratumoral permeability [5,8,9,10].

Quantitative MRI methods, such as dynamic contrast-enhanced (DCE) MRI, are becoming increasingly common in BTB permeability assessment of cancer, as well as for the assessment of permeability responses to treatment options [11,12,13,14,15]. DCE MRI involves the acquisition of a series of T1-weighted (T1-w) images before and after an i.v. bolus injection of gadolinium chelates to capture the contrast agent’s kinetics. A pharmacokinetic (PK) model can then be applied to these dynamic images to estimate several vascular permeability/perfusion parameters based on signal intensity (SI) changes over time [16,17,18,19]. However, conventional PK modeling is complex and time-consuming for a whole DCE MRI scan with multiple dynamic images and thousands of pixels per image.

Deep learning, specifically convolutional neural networks (CNN), has been successfully applied to many image processing applications and can solve complex problems through pattern recognition [20,21]. Researchers have introduced CNNs to facilitate a vast amount of medical imaging research, including image segmentation, detection and classification of malignancies, computer-aided prognosis, image de-noising, and synthetic image generation [22,23,24,25]. Neural networks (e.g., deep learning) can automatically learn about the representations of data during the training process and can be implemented as the surrogate problem solvers of sophisticated and difficult problems. Furthermore, neural networks have been shown to be able to speed up the problem solving process by multiple orders of magnitude. As such, implementing deep learning for PK mathematical modeling in DCE MRI holds great promise for accelerating the prediction of PK parameters in brain tumors without complex and time-consuming mathematical modeling.

Indeed, others have sought to implement deep learning for the prediction of vascular PK parameters based on clinical data. Fang et al. [26] and Ulas et al. [24] have demonstrated the feasibility of implementing 1D and 2D CNNs, respectively, in clinical datasets that does not require conventional PK modeling. These approaches using CNNs could accurately characterize vascular PK parameters and largely speed up the modeling process. Although the CNN is the most commonly employed deep learning approach for imaging processing tasks, other studies have demonstrated alternative deep learning approaches for the prediction of PK parameters without conventional PK modeling in clinical data, including the long short-term memory (LSTM) network [27], the fully connected network (FCN), and the gated recurrent unit (GRU) [28]. However, to the best of our knowledge, there have been no efforts to apply these approaches to small animal permeability research.

Hence, to this end, we have previously developed a deep learning model using a 2D CNN for the efficient generation of PK parameter maps in glioblastoma (GBM)-bearing mice [29]. The CNN was able to accurately predict intratumoral vascular PK parameters of GBM in less than a few seconds, significantly decreasing the average computational time during the PK modeling process. Furthermore, it was shown that the network was transferrable between alternative PK models and parameters. The results and observations from this study indicate that the neural network could perform as precisely as conventional PK models can for predicting permeability and perfusion in small animal brain tumors [29]. However, it is still unknown if the CNN can accurately predict vascular permeability parameters in alternative brain tumor models, as well as predict vascular responses to anti-cancer treatments.

In the present study, to further elucidate and establish the utility of the CNN, we have sought the expansion of our CNN for the assessment of BTB permeability responses to radiotherapy in a breast cancer MDA-MB231-Br (231-Br) BM mouse model. Our data have demonstrated that the model leads to the development of multiple metastases throughout the brain that can be as small as 0.1 mm^3^ (6–8 pixels) on high-resolution T2-weighted MRI images (T2-WI) [5]. A large proportion of these BM lesions retain an intact BTB even at the late stage of the disease, leading to heterogeneity of permeability both between and within lesions. As such, this BM model serves as both a difficult and excellent phenotype for evaluating the expansion of our CNN in the prediction of vascular permeability.

A transfer learning approach was employed to transfer our GBM-trained CNN to BM DCE MRI datasets treated with or without WBRT. Transfer learning is a powerful tool in machine learning that allows the knowledge gained from one problem to be passed on to another related problem [30,31,32]. The application of transfer learning can reduce the training times and can improve the network’s performance and generalization when it is applied to other related problems. Here, we apply transfer learning to transfer the knowledge gained for predicting GBM PK parameters to predicting permeability responses to radiotherapy in BM mice based on DCE MRI (Figure 1). To the best of our knowledge, this is the first study employing the transfer learning of a CNN for the prediction of vascular permeability responses to therapy in an alternative tumor phenotype in either pre-clinical or clinical settings.

## 2. Materials and Methods

### 2.1. GBM and BM Mouse Models

All animal procedures performed were approved by the Wake Forest University Institutional Animal Care and Use Committee. The orthotopic GBM and breast cancer BM mouse model have been described previously [5,29,33,34]. Briefly, human GBM U87 cells (ATCC, Manassas, VA, USA) and MDA-MB-231-BR (231-BR) BM cells (kindly provided by Dr. Steeg, NCI) were cultured in DMEM with 10% FBS, 1% L-Glutamine, and 1% penicillin-streptomycin at 37 °C with 5% CO_2_. Once 80% confluence was achieved, cells were harvested and suspended in serum-free medium. U87 cell suspensions (1.5 × 10^4^ cells in 4 μL serum-free medium) were injected intracranially to the right caudal diencephalon of nude mice (*n* = 6). Under the guidance of a small animal ultrasound imaging platform (Vevo LAZR, FUJIFILM VisualSonics, Inc. Toronto, Canada), 231-BR cell suspensions (2 × 10^5^ cells in 50 μL serum-free medium) were injected directly into the left ventricle of nude mice (*n* = 16). All animals were anesthetized via the inhalation of isoflurane (3% induction; 1.5% maintenance) during the procedures.

### 2.2. Magnetic Resonance Imaging

MR imaging was performed with a small animal Bruker 7T Biospec 70/30 USR scanner (Bruker Biospin, Rheinstetten, Germany). Animals were anesthetized with 3% isoflurane, and the tail vein was catheterized using a 27G butterfly for the bolus injection of Gd-DTPA (Magnevist; Bayer Healthcare). A respiratory bulb was used during image acquisition to monitor the animals’ respiration. MRI was performed using our previously established small animal imaging protocol [5,29]. Briefly, anatomical T2-WI were acquired using a Rapid Imaging with Refocused Echoes (RARE) sequence (TR/TE: 2500/50 msec; Number of Scan Averages (NSA): 8; Echo Train Length (ETL): 8, Field of View (FOV): 22 mm × 22 mm (256 × 256 pixels); Slice Thickness (ST): 1 mm; scan time: 5 min and 22 s). Variable flip angle images were captured for T1 mapping prior to DCE MRI (TR/TE: 100/2.24 ms; NSA: 6; FOV: 22 mm × 22 mm (128 × 128 pixels); ST: 1 mm; Flip Angles (FA): 5, 10, 20, and 35 degrees; scan time: 57 s/FA). DCE MRI was performed on 5 slices using a rapid T1-w FLASH sequence (TR/TE: 43/2.3 ms; FA: 30 degrees; FOV: 22 mm × 22 mm (128 × 128 pixels); NSA: 2; scan time per acquisition: 8 s) acquired during the bolus injection of Gd-DTPA (0.1 mmol/kg, i.v.). Lastly, T1-w contrast-enhanced (T1-CE) imaging was performed using a T1-w RARE sequence (TR/TE: 800/7 ms; ETL: 8; NSA: 8; FOV: 22 mm × 22 mm (256 × 256 pixels); scan time: 2 min and 33 s).

For GBM-bearing mice (*n* = 6), MRI was performed two weeks after tumor implantation. Alternatively, for the BM mice (*n* = 16), MRI was performed three weeks after tumor implantation. A subset of BM mice (*n* = 10) was randomly assigned to either WBRT (*n* = 5) or sham (*n* = 5) treatment groups. Immediately following the initial imaging of BM mice and the confirmation of the development of multifocal BM lesions, WBRT or sham irradiation was conducted for each treatment group in 3 daily doses. Twenty-four hours following the last WBRT dose, MRI was acquired again for both treatment groups to assess permeability/perfusion changes at mid-WBRT. The remaining BM mice (*n* = 6) were used as training data and not included in network testing. A 3D Gaussian filter was used on DCE dynamic data to smooth images prior to the PK modeling process. The co-registration of DCE and variable flip angle images was performed, with T2-weighted images serving as the reference to eliminate motion artifacts in the PK modeling process.

### 2.3. Whole-Brain Radiotherapy

BM-bearing mice were randomly assigned to either WBRT (*n* = 5) or sham (*n* = 5) treatment groups. Following first round of MRI three weeks after tumor implantation and the confirmation of the development of multifocal BM lesions, mice were anesthetized with 3% isoflurane, and the WBRT treatment group received three daily doses of 4 Gy, as described previously [5]. An X-RAD 320 orthovoltage irradiator (Precision X-ray, North Branford, CT) was used to deliver WBRT at 300 kV and a dose rate of 233 cGY/min. A 1 mm Cu (copper) HVL was used to filter the X-ray beam, and a rectangular Lipowitz alloy collimator of 10 × 15 mm was used to align the X-ray beam to the whole brain.

### 2.4. Conventional PK Modeling

All modeling steps were implemented using homemade MATLAB scripts. Prior to conventional PK modeling, T1 mapping was performed to relate the native T1 intensity (T10) of tumor tissue to SI changes during DCE MRI to contrast agent concentration. Co-registered variable flip angle T1 images acquired at angles of 5, 10, 20, and 35 were used to calculate T10 using the method proposed by Brookes et al. [35]. The SI of each flip angle image was stacked into an array for each individual pixel. X and Y coordinates were computed according to the following equations for T10 estimation:(1)$$X=\frac{\mathrm{SI}}{\tan\left(\mathrm{VA}\right)}$$
(2)$$Y=\frac{\mathrm{SI}}{\sin\left(\mathrm{VA}\right)}$$

These coordinates were then linearly fitted to quantify the slope (Sp) and, in turn, tissue T10 values could be estimated by:(3)$$T10=\frac{-\mathrm{Tr}}{\log\left(\mathrm{Sp}\right)}\;,$$
where Tr is the repetition time of the MR scan. All DCE MRI images were temporally resized to 128 × 128 × 5 × 40 (X × Y × slice × time points). Dynamic changes in SI following Gd-DTPA administration were then used to deduct T1 values, which are given by:(4)$$S\left(t\right)={\;S}_{0}\frac{\left(1-e^{\frac{-\mathrm{Tr}}{T1}}\right)*\sin\left(\theta \right)}{1-e^{\frac{-\mathrm{Tr}}{T1}}*\cos\left(\theta \right)}\;,$$
where S_0_ is the relaxed signal before injection of the contrast agent, and θ is the flip angle of the DCE MRI sequence (θ = 30°).

Following T1 mapping, dynamic concentration maps of the contrast within the tissue (C_t_) were computed using the relaxivity of the contrast agent (r1 = 3.11 s^−1^mM^−1^ for Gd-DPTA at 7T) according to the following equation:(5)$$\frac{1}{T1}=\frac{1}{T10}+r1{*C}_{t}\left(t\right)\;,$$

Conventional PK modeling was then performed using the Extended Tofts Model (ETM) according to the following equation:(6)$$C_{t}\left(t\right)={\;V}_{p}{*C}_{p}\left(t\right)+K_{\mathrm{trans}}*\int_{0}^{t}C_{p}\left(t\right){*e}^{-k_{\mathrm{ep}}*\left(t-\tau \right)}d\tau \;,$$
where Vp, Ktrans, and kep are all PK parameters that describe the fractional volume of blood plasma, the transfer rate of contrast agent from the blood plasma to the extravascular extracellular space (EES), and the reverse transfer rate of the contrast agent from the EES to the blood plasma, respectively. C_p_ is a population average bi-exponential arterial input function (AIF). For this study, ETM maps of permeability parameter Ktrans were generated for all DCE imaging data (*n* = 140 slices) and used for network training, as Ktrans is commonly the most reproducible and reliable parameter from conventional PK modeling.

### 2.5. Transfer Learning

All machine learning algorithms were developed and implemented on MATLAB using the deep learning toolkit and an imported Keras library using a 3.7 GHz processor with 16 GB RAM. A 24-layer CNN designed with dual parallel pathways, as described in our previous study, was used for transfer learning [29]. The CNN was originally designed for the prediction of PK parameters in GBM small animals using DCE MRI images. The CNN consists of dual parallel pathways to capture both local and global information. The local pathway consists of 3 convolutional non-dilated layers. The global pathway consists of 3 convolutional layers that were dilated by factors of 2, 4, and 8, respectively. The convolutional layers were designed with a filter size of 4, and there were 128 filters in both local and global pathways. The dual pathways were then concatenated and followed by 4 fully connected convolutional layers of 1024, 512, 128, and 1 hidden node with a filter size of 1. Each convolutional layer was followed by a ReLU activation layer with the exception of the final convolutional layer, which was instead followed by a regression output layer to estimate the output. In the present study, the networks hidden layer’s weights were saved following training with GBM DCE MRI datasets for transfer learning using MATLAB’s deep network designer.

BM DCE MRI datasets (*n* = 140 DCE slices) were concatenated to create a single four-dimensional dataset of size 128 × 128 × 40 × 140. Similarly, BM DCE MRI target ETM PK parameter maps of Ktrans were also concatenated to a single four-dimensional dataset of size 128 × 128 × 1 × 140. DCE dynamic images and target ETM PK parameter maps were segmented to remove peripheral tissue surrounding the brain. The BM DCE MRI dataset was further batch normalized in each plane. As GBM and BM MRI datasets were acquired using the same standard imaging protocol and were subjected to the same PK modeling process, both datasets have the same DCE image input size (128 × 128 × 40), as well as PK image output size (128 × 128 × 1), in the CNN. Hence, domain adaptation in the transfer learning process was not required, as the source (GBM) and target (BM) data distributions/characteristics were the same. A supervised transfer learning technique was then employed through feeding the BM DCE MRI datasets and target ETM maps of Ktrans into the pre-trained GBM CNN. The network was then re-trained to learn about the relationship between SI changes in BM DCE MRI images and the corresponding ETM maps of the permeability parameter, Ktrans.

Training was performed using k-fold cross validation. To this end, imaging datasets from a single WBRT or sham irradiation treated animal (*n* = 10 slices, 5 pre-treatment, and 5 post-treatment) were randomly isolated and removed from the training process and saved for future testing of the network. The remaining DCE imaging datasets (*n* = 130 slices) were then randomly sorted into a training to validation ratio of 80:20. All data were shuffled, and hyper-parameters were adjusted and optimized to achieve the best network performance with minimum error. The following hyper-parameters: a learning rate of 1 × 10^−4^, a maximum number of epochs of 215, and a mini-batch size of 4 using the Adam optimizer, were determined as the optimal tuned parameters for the best prediction accuracy in the transfer learning process. The CNN was then re-trained using the tuned hyper-parameters. Following training, the isolated imaging dataset was then fed directly into the network to predict Ktrans maps. This process was repeated ten times to allow each WBRT or sham irradiation treated animal to serve as the testing data.

### 2.6. Image Processing and Statistical Analysis

Following the training and testing of the transferred network, individual lesions were masked and segmented using newly developed MATLAB scripts. An SI threshold-based segmentation approach was employed on ground truth high-resolution T2-WI of BM lesions. The tumor mean Ktrans, as well as individual pixel Ktrans, were quantified using the segmented masks of BM lesions. All statistical analysis was performed using GraphPad Prism 9.1.

To assess the accuracy of the network for predicting intratumoral BM PK parameters, pixel-wise and lesion-wise comparisons of Ktrans were conducted. Linear regression was applied for statistical correlation and significance. The root-mean-squared error (RMSE) and mean absolute error (MAE), as well as normalized RMSE (nRMSE) and MAE (nMAE) values based on target ETM Ktrans standard distributions (SD), were determined for each individually trained network, as well as for the ensemble of all networks. To assess the accuracy of the network for predicting permeability changes induced by radiotherapy, analysis of variance (ANOVA) was used with Fisher’s LSD multiple comparisons test to determine statistical differences between all treatment groups for both the ETM and CNN.

## 3. Results

The transferred CNN does not require complex mathematical curve fitting algorithms and can efficiently generate PK parameter maps with no human interference or additional data. For a single BM DCE MRI imaging dataset (*n* = 5 slices), the CNN could generate PK parameter maps in 2 s. By comparison, the ETM required 26.2 min to generate PK parameter maps for the same imaging dataset. Furthermore, in contrast to the deep learning approach, conventional PK modeling requires the manual identification of contrast agent arrival on DCE dynamic images to begin the modeling process, the conducting of additional MRI for T1 measurements, and knowledge about the AIF, which further complicates the modeling process. In good agreement with our previous studies with the 231-BR BM model, T2-WI revealed the formation of multifocal hyperintense BM lesions (Figure 2). While some BM lesions were largely enhanced on T1-CE images (green arrows), many of these lesions were found to retain an intact BTB, as evidenced by no enhancement on the T1-CE images (red arrowheads). Furthermore, lesions with a partially intact BTBs were also found, as evidenced by minimal enhancement on the T1-CE images (yellow arrows).

As shown in Figure 2, three representative BM cases tested using the transferred neural network all displayed marked differential permeability levels between individual lesions. Of particular interest, in Case 3, Lesions 1 and 2 (Figure 2b) were both highly enhanced on the T1-CE images and had large increases in the SI on DCE SI time courses, indicating high-permeability Ktrans. Lesion 3 was found to be minimally enhanced on the T1-CE images, with minimal SI changes on its corresponding DCE SI time course, leading to low-permeability Ktrans. Lesion 4 was not enhanced on the T1-CE images and had no increases in the SI on its corresponding DCE SI time course, similarly leading to low permeability Ktrans. Importantly, target conventional ETM maps of permeability parameter Ktrans similarly recapitulated these trends in differential permeability between individual lesions in high agreement to their DCE SI time courses. The transferred neural network similarly recapitulated this trend of intertumoral heterogeneity of Ktrans with high accuracy. Furthermore, the CNN maps depicted some image de-noising relative to the ETM maps, particularly in the surrounding healthy brain region.

Across all the tested DCE imaging datasets (*n* = 100 slices), a total of 276 lesions were identified on the corresponding high-resolution T2-WI. Lesion-wise and pixel-wise (*n* = 11,628 pixels) comparisons of the permeability parameter Ktrans between the ETM and CNN were conducted following an ensemble of all lesions (Figure 3). A significant linear correlation (*p*-value < 0.0001) was found for both the tumor mean Ktrans (R^2^ = 0.54) and the individual pixel comparisons of Ktrans (R^2^ = 0.61). Lesion-wise and pixel-wise comparisons revealed marked heterogeneity both between and within lesions, respectively. As many of these lesions were not enhanced, we further investigated lesion-wise and pixel-wise correlations between the ETM and CNN only for T1-CE enhanced lesions (Figure 3). A significant linear correlation (*p*-value < 0.0001) was found for both the lesion-wise Ktrans (R^2^ = 0.62) and the pixel-wise Ktrans (R^2^ = 0.64).

Clearly, the transferred CNN can decipher both inter- and intratumoral heterogeneity with a high level of correlation with the ETM. We then investigated if the neural network over- or under-predicted the permeability parameter Ktrans in these BM lesions. Bland-Altman plot analysis was performed on all lesions (*n* = 276) for both tumor mean Ktrans and the individual pixel values of Ktrans. As shown in Figure 4, the CNN was found to not over- or under-predict tumor mean Ktrans or individual pixel Ktrans values. To further assess the predictive performance and accuracy of the CNN in BM, RMSE, nRMSE, MAE, and nMAE were quantified for each of the ten trained and tested networks/animals. Similarly, these error terms were also quantified for an ensemble of all networks (Table 1). For each of the ten networks, low RMSE and MAE values were found, eight of which had both an RMSE and MAE smaller than the target ETM SD (nRMSE and nMAE < 1). Importantly, the ensemble analysis of all tested lesions revealed an RMSE value of 6.54 × 10^−4^ and an MAE value of 4.50 × 10^−4^, which was smaller than that of the target ETM SD (nRMSE = 0.638, nMAE = 0.439), indicating that the CNN has a smaller predictive error than the relative SD, as seen on the target ETM maps.

**Figure 4 cancers-15-02703-f004:**
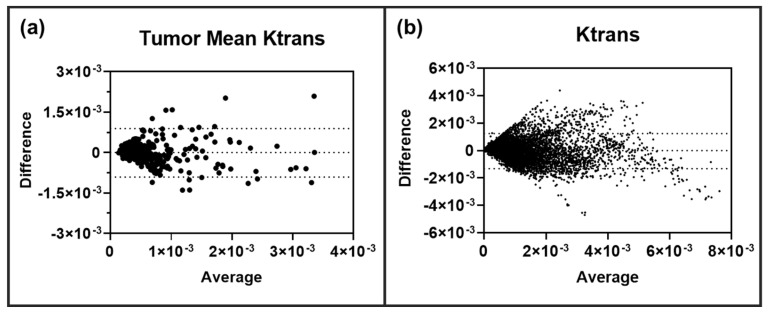
Lesion-wise (**a**) and pixel-wise (**b**) Bland-Altman plot analysis revealed no over or under-prediction of permeability parameter Ktrans in BM lesions (*n* = 276 lesions, 11,628 pixels). 95% limits of agreement are shown as dashed lines.

To assess if the transferred network can decipher permeability changes induced during radiotherapy, we compared the ETM and CNN maps of permeability parameter Ktrans for both pre and post-WBRT and sham irradiation treatments. Lesions that were found to be newly forming or disappeared on the post-treatment images were removed from the analysis to ensure that only lesions affected by either treatment were included. Representative cases of pre- and post-sham irradiation and WBRT treatments are presented in Figure 5a. Anatomical images of the sham treatment group revealed three hyperintense lesions across both the pre- and post-treatment images. Only one of these lesions had a minimal level of enhancement on the pre-treatment T1-CE images (yellow arrow). Post-treatment, both of these lesions became highly enhancing on the T1-CE images (green arrows). All three lesions were found to have low permeability parameter Ktrans on both the ETM and CNN maps pre-treatment. However, post-treatment, two of these lesions had increases in Ktrans on both the ETM and CNN maps, which means that they were in good agreement.

**Table 1 cancers-15-02703-t001:** Predictive performance of the CNN for predicting Ktrans in BM small animal models.

**Metric**	**Animal 1**	**Animal 2**	**Animal 3**	**Animal 4**	**Animal 5**	**Ensemble**	
RMSE	5.70 × 10^−4^	5.39 × 10^−4^	7.73 × 10^−4^	6.50 × 10^−4^	6.82 × 10^−4^		
nRMSE	0.754	0.938	1.21	0.639	0.508	RMSE	6.54 × 10^−4^
MAE	3.83 × 10^−4^	3.79 × 10^−4^	5.20 × 10^−4^	5.03 × 10^−4^	4.89 × 10^−4^		
nMAE	0.507	0.659	0.814	0.495	0.364	nRMSE	0.638
**Metric**	**Animal 6**	**Animal 7**	**Animal 8**	**Animal 9**	**Animal 10**		
RMSE	6.44 × 10^−4^	5.46 × 10^−4^	6.88 × 10^−4^	1.74 × 10^−3^	6.65 × 10^−4^	MAE	4.50 × 10^−4^
nRMSE	0.485	0.457	0.880	2.46	0.595		
MAE	4.52 × 10^−4^	3.95 × 10^−4^	4.42 × 10^−4^	1.48 × 10^−3^	5.07 × 10^−4^	nMAE	0.439
nMAE	0.340	0.331	0.566	2.09	0.453		

The anatomical images of the representative case for the WBRT treatment group (Figure 5a) revealed six hyperintense lesions on the T2-WI. One of these lesions newly formed post-WBRT. The T1-CE images revealed that only one of these lesions had a minimal level of enhancement pre-WBRT (yellow arrow). Following WBRT, the T1-CE images revealed three highly enhanced lesions (green arrows) and two minimally enhanced lesions (yellow arrows), indicating enhanced permeability. All lesions, with the exception of the newly formed lesion, were enhancing post-WBRT. The pre-treatment ETM and CNN maps revealed that all lesions had a low intratumoral Ktrans prior to WBRT. On the post-WBRT images, multiple lesions had increased permeability parameter Ktrans with good correlation with the T1-CE images. Importantly, these trends in permeability changes for both the sham and WBRT treatment groups were recapitulated, with the CNN in good agreement with the ETM.

Clearly, both WBRT and sham irradiation lead to increases in permeability on both ETM and CNN maps (Figure 5a). In order to further quantify and compare the extents to which each treatment increased the permeability parameter Ktrans, tumor mean Ktrans were quantified for all groups for both the ETM and CNN (Figure 5b). The tumor mean Ktrans was found to be increased in the sham radiation group (*n* = 79 lesions) post-treatment, but it did not reach statistical significance for both the ETM and CNN. Alternatively, both the ETM (*p*-value = 0.0022) and CNN (*p*-value = 0.001) revealed significant increases in tumor mean Ktrans in the WBRT group (*n* = 34 lesions) post-treatment. Moreover, post-WBRT lesions were found to have significantly higher tumor mean Ktrans than post-sham irradiation lesions for both the ETM (*p*-value = 0.0067) and the CNN (*p*-value = 0.0454).

**Figure 5 cancers-15-02703-f005:**
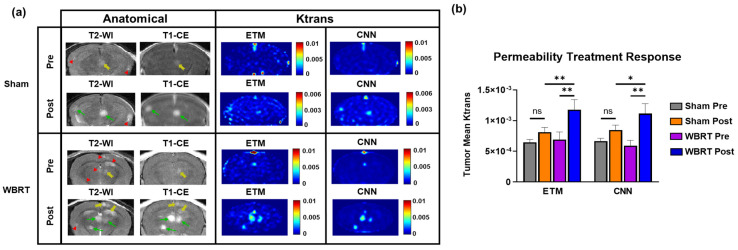
Investigation of permeability changes pre- and post-sham/WBRT. (**a**) Anatomical MR images revealed multifocal lesion development with differential permeability for both sham and WBRT treatments. Green arrows, yellow arrows, and red arrowheads correspond to enhanced, minimally enhanced, and non-enhanced lesions, respectively. Corresponding pre- and post-treatment ETM and CNN maps revealed increases in permeability parameter Ktrans post-treatment, which is in good agreement for both sham and WBRT treatments. (**b**) Quantification of tumor mean Ktrans permeability treatment response. Despite an increase in Ktrans post-treatment, no statistical significance was found between pre- and post-sham irradiation treatment groups for both the ETM and CNN. Significant increases in tumor mean Ktrans were found for the WBRT treatment groups for both the ETM (*p*-value = 0.0022) and CNN (*p*-value = 0.001). Post-WBRT lesions had significantly higher tumor mean Ktrans than post-sham irradiation lesions for both the ETM (*p*-value = 0.0067) and the CNN (*p*-value = 0.0454). Mean ± SEM, * *p* < 0.05, ** *p* < 0.01.

## 4. Discussion

In the present study, we have established a transfer learning approach to using a machine learning model for predicting permeability in a small animal model of GBM to predict radiotherapy treatment responses in a small animal model of BM. To the best of our knowledge, this is the first study on transfer learning employing a CNN for the prediction of vascular permeability responses to therapy. In general, deep learning requires a large amount of data to develop a model with accurate predictions and good generalization. With the limited availability of animal data used in this study, there is a higher risk of the development of a model with inaccurate predictions and an inability to generalize to alternative testing cases. However, despite there being a small number of training data (*n* = 140 slices), we have shown, in the current study, that the CNN can efficiently recapitulate vascular permeability responses in multiple testing cases in a small animal model of BM. This is likely attributable to the fact that each of these training slices comprised multiple individual lesions with thousands of pixels per image. Across all DCE training data, 397 lesions for the network to learn from were identified on corresponding anatomical T2-WI. Similarly, through the implementation of transfer learning, the CNN was provided a more robust and accurate starting point for the network to learn from.

As shown in Figure 2, BM lesions presented with marked differential permeability both between and within lesions. Enhanced (green arrows), minimally enhanced (yellow arrows), and non-enhanced lesions (red arrowheads) were identified on anatomical images, representing an excellent tumor phenotype for the assessment of transfer learning of the CNN. Resulting ETM Ktrans maps generated from DCE SI time courses were in good agreement for permeable, minimally permeable, and impermeable lesions. Importantly, the CNN maps show a good match to their corresponding ETM maps for BM and to recapitulate these trends in differential permeability between individual lesions with a high level of accuracy. Furthermore, the CNN maps were shown to successfully recapitulate both the intertumoral and intratumoral heterogeneity of BM lesions with significantly strong correlations to the ETM maps (Figure 3). An ensemble across all tested sham and WBRT-treated animals revealed that the network had a lower RMSE (RMSE = 6.54 × 10^−4^) relative to the ETM intratumoral SD (nRMSE = 0.638). Similarly, the ensemble analysis revealed that the CNN had a lower MAE (MAE = 4.50 × 10^−4^) relative to the ETM SD (nMAE = 0.439), indicating the ability of the transferred network to generate Ktrans with a minimal error (Table 1).

In our previous study, we tested the aforementioned GBM trained neural network using BM mice without transfer learning [29]. Interestingly, Bland-Altman plot analysis revealed that the CNN without transfer learning slightly over-predicted the intratumoral permeability parameter Ktrans generated by the ETM. As shown in Figure 4 in the current study, the transferred network did not over- or under-predict the permeability parameter Ktrans in BM lesions. Via the application of transfer learning, the network was able to better generalize to DCE MRI datasets of BM lesions with an enhanced predictive accuracy of PK parameters. Hence, in order for the CNN to have accurate predictions of permeability PK parameters for a specific brain tumor phenotype of interest, it should be trained using images of the target brain tumor model.

To this end, we believe that transfer learning is a robust machine learning approach that can reduce the training times, tolerate a lack of training data, and improve both the network performance and generalization when it is applied to other brain tumor phenotypes. We anticipate that the network could not only be transferred to alternative brain tumor types, but also to other cancer anatomical sites. Although further studies are warranted, the CNN could also transfer to clinical DCE MRI studies. Furthermore, this approach could translate to alternative quantitative imaging techniques beyond DCE MRI via transfer learning, including quantitative apparent diffusion coefficient (ADC) mapping in diffusion-weighted (DW) MRI and perfusion metrics, such as cerebral blood volume (CBV), from dynamic susceptibility contrast (DSC) MRI.

As shown in this study, the transfer learning approach has the ability to generate PK parameter maps that are structurally similar and highly correlated with conventional PK model maps with a minimal error and negligible over-/under-prediction. In line with our previous study, the transferred network significantly reduces computational times in comparison to those of the ETM [29]. For a single BM DCE MRI imaging dataset (*n* = 5 slices), the ETM required 26.2 min to generate PK parameter maps. In contrast, the CNN only required 2 s to generate PK parameter maps from the same BM DCE MRI imaging dataset. The machine learning approach negates the need for the manual identification of contrast agent arrival, additional MRI acquisitions for T1 measurements, complex curve fitting, and knowledge about the AIF, all of which are required for conventional PK modeling and can complicate the modeling process. Furthermore, the CNN maps depicted some image de-noising relative to the ETM maps, particularly in the surrounding healthy brain region. CNN approaches have commonly been implemented for noise removal in computer vision tasks [36]. Noise removal in PK parameter maps from DCE MRI can help to improve the image quality and allow the ease of interpretation of permeability and perfusion kinetics potentially improving prognosis and treatment planning.

DCE MRI is becoming increasingly common in the assessment of BTB permeability responses to treatment options. In the present study, we have investigated the ability of the proposed neural network to predict permeability changes in BM following a WBRT treatment. Permeability parameter Ktrans using the ETM was found to significantly increase in post-WBRT tumors relative to both pre-WBRT, as well as post-sham irradiation (Figure 5), indicating the ability of WBRT to disrupt the BTB. This finding supports the potential advantage of concomitant chemotherapeutic agents with WBRT in BM patients. It is worth noting that this increased intratumoral permeability may be only applicable to other small-molecule drugs of a similar molecular weight to that of the used MRI contrast agent, Gd-DTPA [5,9]. Future studies of alternative contrast agents, such as macromolecular contrast agents or albumin labeled contrast agents, may allow us to predict the permeability of the BTB to large therapeutic antibodies.

Importantly, it was shown that the transferred network accurately predicted the permeability parameter Ktrans in both the sham irradiation and WBRT treatment groups (Figure 5). The network similarly revealed a significant increase in Ktrans for post-WBRT tumors relative to those of both the pre-WBRT and post-sham irradiation treatment groups, which is in good agreement with the ETM. This finding supports the use of the proposed neural network as an efficient and accurate tool for PK modeling in the assessment of BTB permeability responses to treatment. To this end, it would be interesting to investigate the utility of the deep learning approach to investigate permeability changes due to radiotherapy treatment in clinical brain tumors that could provide further insight into the potential advantage and timing of concomitant chemotherapeutic agents.

## 5. Conclusions

The proposed CNN allows the rapid estimation of brain tumor PK parameters without complex PK modeling. As shown in this study, by employing transfer learning, the CNN can be used to study alternative small animal brain tumor models. Furthermore, the proposed CNN can serve as an efficient and accurate tool for characterizing permeability treatment responses in small animal brain tumor research. The results and observations from this study support the use for transfer learning for improved network performance and generalization in the prediction of vascular permeability changes following treatment. Future studies investigating the ability of the CNN to predict vascular permeability changes due to radiotherapy in clinical brain tumors, transfer to alternative cancer anatomical sites, as well as transfer to different quantitative imaging modalities will further establish the networks utility. Similarly, as permeability and perfusion are three-dimensional, applying a three-dimensional CNN could enhance the predictive performance of the deep learning approach. Furthermore, with the emergence of more sophisticated deep learning methods, such as the generative adversarial network (GAN), future studies are warranted of potential alternative deep learning approaches that may enhance the prediction accuracy of vascular permeability.

## Figures and Tables

**Figure 1 cancers-15-02703-f001:**
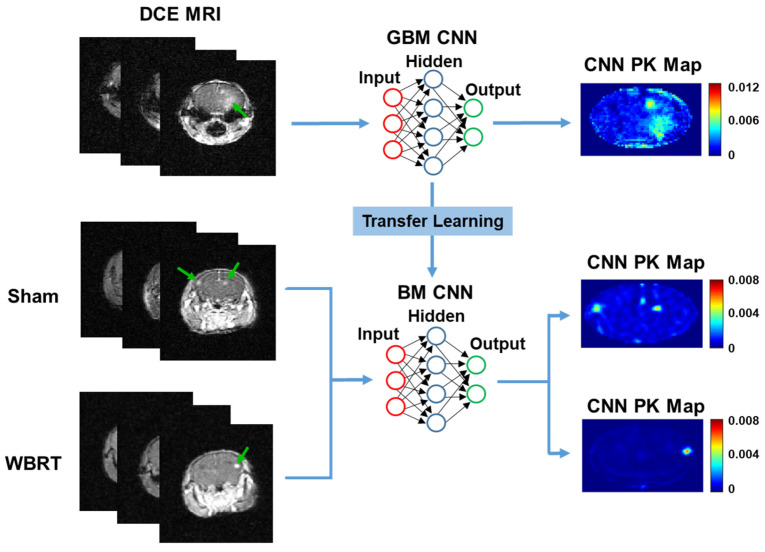
Pathway for transfer learning. A pre-trained GBM CNN hidden layer’s weights were saved, and the network was re-trained using BM DCE MRI datasets. BM mice were treated with either sham irradiation or WBRT and used for the training/testing of the neural network. Green arrows indicate enhanced tumors on DCE MRI.

**Figure 2 cancers-15-02703-f002:**
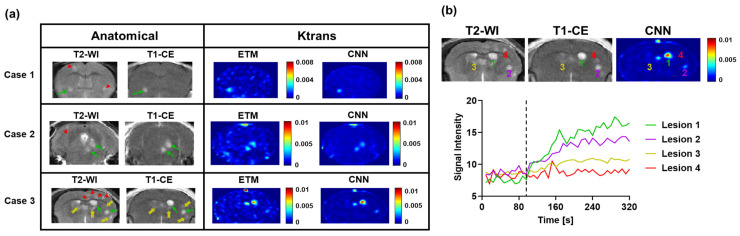
(**a**) Three representative cases of BM imaging datasets tested using the transferred neural network. Anatomical MR images revealed the development of multifocal lesions, many of which retained a partially or fully intact BTB. Green arrows, yellow arrows, and red arrowheads correspond to those enhanced, minimally enhanced, and non-enhanced lesions, respectively. Corresponding ETM and CNN maps of permeability parameter Ktrans display marked differential permeability between lesions with good agreement. (**b**) Representative MR images of Case 3 and corresponding DCE SI changes over time for four lesions. The black dashed line represents the contrast agent arrival. Lesions 1 and 2 displayed high-level permeability and moderate-level permeability, respectively. Lesions 3 and 4 were minimally permeable and impermeable, respectively.

**Figure 3 cancers-15-02703-f003:**
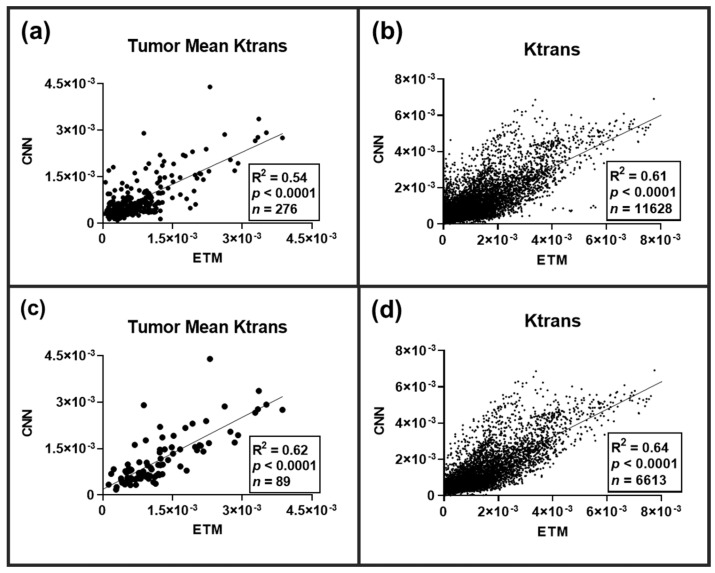
Linear regression analysis plots of BM permeability parameter Ktrans between the ETM and CNN. (**a**) Lesion-wise comparison across all lesions (*n* = 276) revealed a significant linear correlation of R^2^ = 0.54 (*p*-value < 0.0001). (**b**) Pixel-wise comparison across all lesions (*n* = 11,628) similarly revealed significant linear correlation of R^2^ = 0.61 (*p*-value < 0.0001). Lesion-wise (**c**) and pixel-wise (**d**) comparison of T1-CE enhanced lesions (*n* = 89 lesions, 6613 pixels) revealed significant linear correlations of R^2^ = 0.62 and 0.61, respectively (*p*-value < 0.0001).

## Data Availability

All data are available upon request from the corresponding author.

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
