# Peer review of "Transfer Learning Approach to Vascular Permeability Changes in Brain Metastasis Post-Whole-Brain Radiotherapy"

_cancers, 2023, doi:10.3390/cancers15102703_

Round 1
Reviewer 1 Report
This manuscript proposed a novel deep learning method for predicting BM permeability, where CNN and transfer learning were employed for the task of interest. First, a CNN model was trained based on GBM. Then, the trained network was transferred to the predictor applied to whole brain radiotherapy (WBRT) treated brain metastasis (BM) mice based on transfer learning. The performance of the proposed method has been validated using experimental dataset, with satisfactory results. Overall, the topic of this research is interesting, and the manuscript was well organised and written. The detailed comments are summarised as follows.
1. The main innovation and contribution of this research should be clearly clarified in abstract and introduction.
2. Please broaden and update literature review on CNN or deep learning to demonstrate its excellent capacity to resolve practical problems. E.g. Torsional capacity evaluation of RC beams using an improved bird swarm algorithm optimised 2D convolutional neural network; Automated damage diagnosis of concrete jack arch beam using optimized deep stacked autoencoders and multi-sensor fusion.
3. The performance of trained CNN model is heavily dependent on the setting of hyperparameters. How did the authors tune/optimise the network parameters to achieve the best prediction accuracy in this research?
4. How about the characterists of two datasets? If they are not the same, domain adaptation should be considered by adding a discrepancy term in the loss function of CNN model.
5. This study only considers the R-squared and RMSE as evaluation metrics. More metrics should be used for a comprehensive evaluation.
6. A comparison with other models from literature is necessary to prove the superiority of proposed model.
7. More future research should be included in conclusion part.
Reviewer 2 Report
General comments:
This is an interesting and well written paper that demonstrates the feasibility of transfer learning to be employed as a reliable machine learning tool, with a particular example of vascular permeability variations in brain mets after whole brain radiotherapy.
The paper is comprehensive, and the scientific approach is sound.
I only have some minor issues to be addressed by the authors:
(1) Please add to the Discussion section a paragraph on the study limitations. For instance – the pre-clinical nature of your study, the small number of samples (mice) used in each test group which might influence the statistical power of the data.
(2) The authors should also comment on the clinical transferability of the results. The statement in Discussion (lines 411-412) whereby ‘This finding supports the potential advantage of concomitant chemotherapeutic agents with WBRT in BM patients’ requires a comment on the clinical translation of your results.
Minor comments:
Lines 26-27 (abstract) – ‘treated with out without…’-please correct (I assume is ‘with or without’).
Line 77 - replace ‘permeably’ with ‘permeability’
Line 392 – replace ‘negligent’ with ‘negligible’
Lines 233, 248, 398 – what do you mean by ‘contrast agent arrival’?
Line 403 – replace ‘nosie’ with ‘noise’
Overall there are no issues, some typos require corrections as indicated in the comments.
Reviewer 3 Report
I am really grateful to review this manuscript. In my opinion, this manuscript can be published once some revision is done successfully. This study used transfer learning from the pre-trained glioblastoma (GBM) convolutional neural network (CNN) to its brain metastasis (BM) counterpart for mice with sham irradiation or whole brain radiotherapy. The BM CNN registered the legion-wise and pixel-wise R2 values of 0.54-0.64 for the transfer rates of the contrast agent (Ktrans) with respect to the extended Tofts model. The root mean squared error of the former was smaller than that of the latter as well, i.e., 0.638. I would argue that this is a great achievement. But I would like to ask the authors to give a brief description of the CNN in the section of Methods, which is expected to aid in the better understanding of readers.
Minor editing of English language required.
